# Hyperferritinemia—A Clinical Overview

**DOI:** 10.3390/jcm10092008

**Published:** 2021-05-07

**Authors:** Miriam Sandnes, Rune J. Ulvik, Marta Vorland, Håkon Reikvam

**Affiliations:** 1Department of Clinical Science, University of Bergen, 5021 Bergen, Norway; miriam.sandnes@uib.no (M.S.); rune.ulvik@uib.no (R.J.U.); 2Department of Cancer Genomics, Haukeland University Hospital, 5021 Bergen, Norway; marta.vorland@helse-bergen.no; 3Department of Medicine, Haukeland University Hospital, 5021 Bergen, Norway

**Keywords:** ferritin, iron, inflammation, hemochromatosis

## Abstract

Ferritin is one of the most frequently requested laboratory tests in primary and secondary care, and levels often deviate from reference ranges. Serving as an indirect marker for total body iron stores, low ferritin is highly specific for iron deficiency. Hyperferritinemia is, however, a non-specific finding, which is frequently overlooked in general practice. In routine medical practice, only 10% of cases are related to an iron overload, whilst the rest is seen as a result of acute phase reactions and reactive increases in ferritin due to underlying conditions. Differentiation of the presence or absence of an associated iron overload upon hyperferritinemia is essential, although often proves to be complex. In this review, we have performed a review of a selection of the literature based on the authors’ own experiences and assessments in accordance with international recommendations and guidelines. We address the biology, etiology, and epidemiology of hyperferritinemia. Finally, an algorithm for the diagnostic workup and management of hyperferritinemia is proposed, and general principles regarding the treatment of iron overload are discussed.

## 1. Introduction

Ferritin is one of the most commonly requested laboratory tests in general and secondary care, and levels deviating from reference ranges are a frequent finding [1]. Ascribed to its proportionality to total body iron stores, ferritin function is an indirect marker of iron status [2]. When concurrent inflammation is absent, ferritin has proven to be a highly specific and sensitive parameter for the diagnosis of iron deficiency [3,4]. High ferritin, hyperferritinemia, may indicate increased iron stores, but is more commonly seen upon acute phase reactions and as a result of ferritin being released from damaged cells such as hepatocytes in liver disease [5]. It may also be the result of increased synthesis and/or increased cellular secretion of ferritin upon various stimuli such as cytokines, oxidants, hypoxia, oncogenes, and growth factors [6].

Ferritin reference ranges may vary according to the analytical assay being used, although upper cut off is typically set to 200 μg/L in women and 300 μg/L in men [1,7]. In a prospective Danish population-based study, ferritin proved to be a strong predictor of premature death in the general population. Subjects with a baseline ferritin ≥200 μg/L were found to have increased risk of cause-specific mortality due to cancer, endocrinological disease, and cardiovascular disease, as well as increased total mortality compared to those with levels <200 μg/L. The study furthermore found a stepwise increase of this risk upon stepwise increases in ferritin, with the highest cumulative risk seen upon levels ≥600 μg/L [8].

Clinical interpretation of hyperferritinemia often proves to be complex, and ferritin >1000 μg/L is regarded as a non-specific marker of pathology. General practitioners seem somewhat unfamiliar with the appropriate management of hyperferritinemia, as >50% of primary care patients presenting with ferritin levels of such magnitude, and without any obvious clinical reason, are not referred to secondary care nor offered any further investigation [9]. Based on the wide etiological spectrum, hyperferritinemia should prompt for further investigation through clinical examination and additional laboratory tests when the cause remains unknown [1,7].

In this overview, we discuss the biology, etiology, and clinical epidemiology of hyperferritinemia, based on a non-systematic selection of the literature, including international recommendations and guidelines and the authors’ own experiences. Understanding of the fundamental aspects of iron metabolism and the mechanisms promoting elevations in ferritin levels is a prerequisite for rationally dealing with hyperferritinemia in clinical practice, and key points regarding this matter are therefore briefly presented below. Finally, we propose an algorithm for the further investigation and management of hyperferritinemia before general principles regarding the treatment of iron overload are addressed.

## 2. Ferritin and Iron Homeostasis

Ferritin is mostly found as a cytosolic protein, where it plays an important role in the storage of intracellular iron, sequestering up to 4500 Fe^3+^ atoms per molecule. It is a 24-subunit molecule composed of two structurally distinct subunits, the light-chain (molecular weight 19 kilodalton) and the heavy-chain (molecular weight 21 kilodalton) [10]. Ferritin levels are upregulated and matched to intracellular iron levels through the activated translation of heavy- and light-chain mRNA upon high intracellular iron levels. The opposite is true upon low intracellular iron levels, resulting in prevented translation through specific sequences of heavy- and light-chain mRNA that block the recruitment of ribosomal subunits, as illustrated in Figure 1 [11].

Within cells, limited amounts of free iron are found in a labile pool, which is biologically active in metabolism, although toxic if present in excess. By capturing and buffering iron, ferritin plays a key role in maintaining iron homeostasis [12]. Through its ferroxidase activity, heavy-chain ferritin subunits transform ferrous iron (Fe^2+^) into the less toxic ferric state (Fe^3+^). Homozygous murine knockouts of heavy-chain ferritin were found to be embryonically lethal, illustrating its importance in cellular defense, the detoxification of iron, and in maintaining iron bioavailability [13].

Through mechanisms that are not fully understood, small quantities of ferritin emerge into the serum. Controversial results regarding the subunit composition and iron content of ferritin found in serum are reported [14,15,16]. It is, however, assumed to be relatively iron-poor and almost entirely made up of light-chain subunits [6,17]. In the normal state, 50–80% of serum ferritin is glycosylated as a result of release from macrophages of the reticuloendothelial system (RES), and to some extent hepatocytes [1,18,19]. In certain hereditary hyperferritinemic states, glycosylation is almost 100% [20]. This supports a regulated release mechanism, whilst a higher percentage of non-glycosylated ferritin upon liver necrosis suggests hyperferritinemia to be the result of passive release, mainly due to cellular damage [1].

A schematic overview of the human iron metabolism is shown in Figure 2. The uptake of heme and non-heme iron takes place through the enterocytes of the proximal duodenum. Once inside the enterocyte, iron has two possible pathways—one portion remains intracellularly for use or for storage, while the rest is transported across the basolateral membrane through the iron exporter ferroportin, upon which it subsequently binds to transferrin. Effective efflux of iron through the basolateral membrane requires iron to be oxidized. This is mainly facilitated by the ferroxidase activity of plasma ceruloplasmin, which is synthesized in the liver, and its membrane-bound intestinal homolog hephaestin [21].

Transferrin is capable of sequestering only two iron atoms, and is also predominantly synthesized in the liver [22]. It is the major serum iron-binding protein, keeping iron biologically accessible in an aqueous environment for delivery to cells through the transferrin receptor 1 (TfR1), allowing Fe^2+^ atoms to be internalized through receptor-mediated endocytosis. Almost all tissue and cells express TfR1. Cells with increased iron demand have a particularly high expression, including rapidly dividing cells such as activated lymphocytes, as well as erythroblasts, which depend on iron delivery for hemoglobin production in erythropoiesis [21,23].

Ferroportin is also expressed in macrophages of the RES, involved in the recycling of iron from the hemoglobin of old red blood cells (RBCs) [24]. Irrespective of levels, iron is eliminated at a basal rate through the desquamation of skin and intestinal epithelium, and through blood loss in fertile women [25]. Both intracellular ferritin and the intestinal absorption of iron are important regulators of iron homeostasis, due to a lack of active physiological iron excretion mechanisms. Systemic regulation of metabolism and the modulation of availability to meet iron needs are predominantly mediated through the peptide hormone hepcidin and the hepcidin–ferroportin axis (Figure 2).

Hepcidin is mainly synthesized by hepatocytes, and facilitates synchronized iron metabolism between various organs, protecting the body against iron overload [26]. It is considered a negative regulator of serum iron levels as it mediates ubiquitin-mediated degradation of ferroportin, which shuts off export of iron to plasma. Iron is consequently retained in the intestinal epithelium, while iron recovery from senescent RBCs is interrupted by inhibited release from RES macrophages [21]. Hepcidin also facilitates a reduction in iron uptake by enterocytes through inhibited transcription and increased degradation of the intestinal iron transporter divalent metal transporter 1 (DMT1) [27,28]. Dysregulation of the hepcidin–ferroportin axis, promoting uncontrolled intestinal absorption and uninhibited cellular export of iron, is a characteristic feature in various iron-loading conditions exhibiting high serum iron levels [29,30], and is later discussed.

Finally, whether hyperferritinemia caused by disease processes has a causal role or a role in cellular protection is not fully established yet. Apart from ferritin’s iron-storing properties, the biological purpose of it remains partly unknown. Nevertheless, increasing evidence of ferritin subsuming additional physiological roles is emerging, and it is now suggested as molecule contributing to inflammation, iron delivery and angiogenesis, as well as cell signaling, proliferation and differentiation [18,21]. Increased understanding of these mechanisms might grant additional insight into the pathophysiology of iron overload, cancer, and inflammation, which may contribute to the development of novel therapeutic targets.

## 3. Etiology

Underlying conditions upon hyperferritinemia, with and without an associated iron overload, are summed up in Table 1. Transferrin saturation is a useful parameter for the distinction of the presence or absence of an iron overload upon hyperferritinemia. It is a calculated value reflecting the proportion of iron-binding sites on transferrin that are occupied [7]. Whilst a normal transferrin saturation usually excludes pathologically increased iron absorption, it does not necessarily exclude the presence of an iron overload [1,31,32,33]. Increases in transferrin saturation are not always equivalent to an iron overload either, and the interpretation of this parameter therefore requires careful considerations [29].

This differentiation of hyperferritinemia is essential, as the management, treatment, and prognosis greatly differ for the two entities. Estimates show that only 10% of clinical cases of hyperferritinemia in routine medical practice are associated with an iron overload. For the rest, one of the following underlying causes attributing to a reactive increase are usually identified: inflammation, metabolic syndrome, chronic alcohol consumption, cellular damage, and malignancy [34,35,36].

### 3.1. Hyperferritinemia without Iron Overload

All forms of inflammation, regardless of its cause, may elevate ferritin levels, and are thus recognized as an acute phase reactant. Pro-inflammatory cytokines stimulate the synthesis of ferritin and hepcidin, leading to hyperferritinemia, iron retention in macrophages, and less iron being available for erythropoiesis due to the redistribution of body iron from RBCs to tissue cells [37]. This is commonly known as anemia of inflammation, which is part of the innate immune defense against invading pathogens and tumor progression [38,39].

Prostaglandins involved in inflammatory and febrile responses, as well as viral replication, have also been demonstrated to induce light-chain ferritin synthesis [6]. Inflammation may induce apoptotic pathways through cytokines, contributing to cellular damage with the concurrent release of ferritin [40], and mechanisms of reactive increases in ferritin thus work synergistically and cannot solely explain the cause of hyperferritinemia. An example of this is seen in autoimmune diseases and cancer, with increased synthesis of ferritin due to inflammation as well as induced cellular damage with the release of ferritin [6,18,40].

If common clinical conditions can be excluded, clinicians must recognize hyperferritinemia as a clue to various autoimmune, inflammatory and genetic disorders. Rare immune-mediated conditions such as hemophagocytic lymphohistiocytosis (HLH), where monocytes and macrophages seem to play a vital role through the production and release of ferritin, may cause extremely elevated ferritin levels [41]. It has been proposed that these high levels are not only a product of inflammation, but also play a pathogenic role themselves, causing extreme expression of additional inflammatory mediators, known as a cytokine storm [41]. 

Trends in ferritin have proven to be a prognostic marker in pediatric HLH patients, as a decrease <50% within 10 weeks after diagnosis showed a 17 times higher mortality risk compared to those with a decrease ≥96% [42]. Ferritin >10,000 μg/L is highly specific and sensitive for the diagnosis of HLH in pediatric patients [43]. This is, however, not true in the adult patient population, as a variety of conditions such as chronic kidney disease, infection, and hematological malignancies may exhibit ferritin levels >50,000 μg/L [44].

Coronavirus disease 2019 (COVID-19) emerged as a pandemic in 2020 and is associated with a hyperactive immune response upon severe disease, which correlates with a high degree of morbidity and mortality. All patients with severe COVID-19 should be screened for hyperinflammation using laboratory parameters such as ferritin [45], which has proven to be a prognostic marker and an indicator of inflammation in these patients [46]. Extreme hyperferritinemia with a cytokine profile similar to that seen in secondary HLH is reported in a subgroup of patients. Serial measurements of ferritin may help monitor this hyperinflammatory state and treatment response, as well as predict worsening and mortality in hospitalized COVID-19 patients [47].

Hematological and various solid malignancies may also induce elevations in ferritin, thought to be the result of both inflammation and cytolysis [1,18], and ferritin levels >1000 μg/L are often seen upon metastatic cancer [48]. No studies have provided evidence of ferritin contributing to the etiology of cancer, rather than merely being a marker for the presence of it. However, several pro-oncogenic functions of ferritin are suggested, as free iron released from ferritin and hemosiderin may potentially catalyze the formation of powerful oxidizing agents capable of promoting lipid peroxidation, mutagenesis, DNA strand break, the activation of oncogenes, and inhibition of tumor suppressors [18].

Cellular damage may induce great rises in ferritin levels. With the liver being the major storage organ of iron, ferritin may reach >10,000 μg/L upon acute and chronic hepatopathy, including alcoholic liver disease and non-alcoholic fatty liver disease (NAFLD), and is partly as a result of cellular damage [7,49]. It should be noted that low serum transferrin due to the impaired synthetic function of the liver upon chronic liver disease may be misleading in the diagnostic workup, as this potentially results in an elevated transferrin saturation, even in the absence of an iron overload [7,50].

An isolated ferritin <1000 μg/L due to daily alcohol consumption is a common presentation [7], seen in up to 40–70% of chronic alcoholics [51]. Experimental models have shown that alcohol promotes the direct stimulation of ferritin synthesis and suppresses hepatic hepcidin expression [52,53], which may account for the linear correlation between alcohol intake and serum iron indices in alcoholics [54,55]. Clinical trials have furthermore shown increases in serum iron and ferritin to be greater upon beer consumption compared to the consumption of wine and spirits [56]. Functioning as both a diagnostic and therapeutic test for alcohol-induced hyperferritinemia, withdrawal results in a rapid decline in ferritin [57].

Liver steatosis and insulin resistance is a frequent finding in patients referred for suspected hemochromatosis on the basis of hyperferritinemia [58]. Insulin has been implicated to induce ferritin synthesis at mRNA level in experimental models, proposing a novel explanation for the hyperferritinemia commonly seen upon insulin resistance [6]. The association between ferritin and metabolic syndrome has been suggested being mainly mediated by undiagnosed NAFLD, which is considered to be the hepatic manifestation of metabolic syndrome. 

Although subclinical inflammation commonly seen upon metabolic syndrome and NAFLD also influence the described association, C-reactive protein (CRP) did not correlate with ferritin levels in the general population or in patients with metabolic syndrome. Studies have also found ferritin in healthy individuals to remain positively correlated with blood glucose and insulin resistance after adjusting for CRP levels [59,60]. These results show that inflammation is unlikely to play a predominant role in determining ferritin levels in these patients [61,62,63,64], and that the pathogenesis is also related to steatosis, hyperinsulinemia, and cellular damage [65]. NAFLD and its association with perturbations of iron homeostasis is also becoming increasingly evident [64], and is discussed later.

Rare genetic causes of hyperferritinemia without an associated iron overload include hereditary hyperferritinemia cataract syndrome (HHCS), caused by variants in the ferritin light-chain gene (*FTL* gene). After common causes have been ruled out, such genetic variants may help explain rare clinical cases of unexpected and isolated hyperferritinemia. HHCS is characterized by an unleashed ferritin light-chain synthesis, ultimately causing bilateral cataracts at an early age due to the deposition of ferritin in the ocular lenses [66]. Variants in the *FTL* gene causing an isolated hyperferritinemia without any symptoms are also reported and are referred to as benign hyperferritinemia [20,67].

Another rare genetic cause related to hyperferritinemia with a normal transferrin saturation is Gaucher disease. It is the most common lysosomal storage disorder and is frequently underdiagnosed, even when all clinical symptoms are present. Gaucher disease should be considered when patients present with unexplained cytopenia and hepatosplenomegaly [68,69].

### 3.2. Hyperferritinemia with Iron Overload

Iron overloading diseases are classified as (1) primary when caused by an inherited defect in the regulation of iron balance and (2) secondary when acquired as a result of underlying congenital or acquired conditions [70]. During iron-loading conditions where the iron binding capacity of transferrin is exceeded and a high transferrin saturation is observed, non-transferrin bound iron (NTBI) enters circulation. NTBI predominantly enters hepatocytes, but also the parenchymal cells of the heart, pancreas, thyroid, and central nervous system. This diffuse distribution of iron is associated with organ dysfunction through cell death and complications such as fibrosis, atherosclerosis, and carcinogenesis [23,30,71].

#### 3.2.1. Primary Iron Overload

Primary iron overload, synonymous with hereditary hemochromatosis (HH), is, for all practical purposes, the leading cause of severe iron overload. HH is frequently unrecognized in primary care due to a preclinical phase of years [72]. Associated complications such as liver fibrosis and endocrinological disease are serious and potentially preventable, making timely diagnosis and treatment important [8,33]. HH is a result of the previously mentioned dysfunction of the hepcidin–ferroportin regulatory axis, leading to increased serum iron, iron depleted macrophages, accelerated dietary iron absorption, and, finally, parenchymal iron overload in the majority of cases [73]. A schematic overview of the molecular basis of HH due to impaired hepcidin synthesis is shown in Figure 3.

Approximately 82–90% of clinical disease in Caucasian HH patients is related to homozygosity for the missense variant C282Y of the homeostatic iron regulator gene (*HFE* gene), referred to as hemochromatosis type 1 or *HFE* hemochromatosis [23,29,74,75]. It is the most common monogenic disorder in northern European populations, with a prevalence of 0.5% [31,35]. *HFE* hemochromatosis is, however, far less common, and practically absent, among individuals of Asian and Hispanic heritage [32,33]. Penetrance is incomplete, with phenotypic expression in the form of elevated ferritin levels occurring in 80% of male and 50% of female C282Y homozygotes. Furthermore, proportion of C282Y homozygotes with definite disease manifestations such as liver disease or arthritis greatly varies between genders, and is significantly lower in women (1%) than in men (28%) [1,76].

While C282Y is considered the “major” *HFE* hemochromatosis-associated variant, H63D is considered to be the “minor” variant, which seldom causes significant iron overload, even when it is present in compound heterozygosity with C282Y [31]. Phenotypic penetrance in C282Y/H63D compound heterozygotes is approximately 2–5% [77], although only 0.5–2% of these develop clinical signs of iron overload [35,78,79]. This overall low disease penetrance of *HFE* hemochromatosis is partly the rationale for recommendations stating that general population screening for *HFE* hemochromatosis through genetic testing is *not* recommended [31,36,80]. 

Iron overload may be related to the C282Y or H63D variant in compound heterozygosity with rarer *HFE* variants. It is suggested that these variants are the most frequent cause of iron overload in cases of non-C282Y homozygous *HFE* hemochromatosis [81]. When such rare *HFE* variants are not involved, accompanied environmental or host risk factors are identified in practically all cases of iron overload in patients with *HFE* hemochromatosis related to C282Y or H63D heterozygosity, C282Y/H63D compound heterozygosity, or H63D homozygosity.

This includes comorbidities such as metabolic syndrome, hepatopathy, diabetes, or chronic alcohol abuse, ultimately contributing to iron homeostasis dysfunction more than the genetic abnormality itself [1,3,31,77,80]. When an unexpected severe iron overload is detected in these patients, further investigation for possible digenic inheritance or variants in other contributing genes might be of interest [1,29], as this may affect phenotypic and clinical expression [31,82]. It should also be noted that transferrin saturation predicts hepatic iron overload (HIO) better in C282Y homozygotes than in non-C282Y homozygotes and non-*HFE* iron overload (type 2–4 HH). Accordingly, iron overload may be proven in patients with elevated ferritin, despite them having normal transferrin saturation [7,31].

In areas such as Southern Europe and Asia, non-*HFE* HH represents a larger proportion of clinical HH cases [83]. This is a heterogenous group caused by genetic variants unrelated to the *HFE* gene, and genetic testing for these is largely unavailable [31]. Type 2 HH, known as juvenile hemochromatosis, is caused by variants of the hepcidin (*HAMP*) and hemojuvelin (*HJV*) genes, while type 3 HH is related to variants of the TfR2 (*TFR2*) gene.

Finally, type 4 HH is seen as a result of ferroportin 1 (*FPN1*) genetic variants [84]. It is distinct from type 1-3 HH as it is not related to the impaired synthesis of hepcidin, but rather the impaired function of ferroportin (type 4A HH), known as ferroportin disease, or ferroportin being resistant to hepcidin stimulus (type 4B HH). While the latter results in a clinical phenotype similar to that of *HFE* hemochromatosis, a more distinct clinical picture is seen in ferroportin disease, with iron retention mainly being localized to the macrophages of the spleen, liver, and spine, with marginal delivery of iron to circulating transferrin, resulting in a normal or low transferrin saturation (Figure 4). Such a triad of iron retention, shown by abdominal magnetic resonance imaging (MRI), enables the characterization and diagnosis of ferroportin disease. It should also be noted that the clinical expression of ferroportin disease tends to be milder than that of other forms of HH. A non-aggressive phlebotomy regimen is thus recommended in these patients due to the risk of anemia [85]. When anemia occurs during repeated phlebotomies in patients with primary iron overload, ferroportin disease should be considered [86].

Type 4A HH is highly prevalent in African populations, making *FPN1* the gene most frequently associated with hereditary hyperferritinemia in this area. Numerous variants associated with ferroportin disease have also been identified in people of Thai, Japanese, European, and French-Canadian heritage [85].

Extremely rare causes of primary iron overload include genetic defects of the iron metabolism caused by variants in the ceruloplasmin and transferrin gene. When ceruloplasmin activity is impaired (hypoceruloplasminemia) or absent (aceruloplasminemia) due to such genetic variants, marked iron overload ultimately develops as a result of cellular iron retention. Brain iron accumulation in aceruloplasminemia, which causes progressive neurological symptoms, makes this disorder unique among other systemic iron overload syndromes [87]. Hypotransferrinemia with low transferrin concentrations also causes systemic iron overload and is characterized by impaired erythropoiesis and microcytic anemia due to the marked reduction in iron delivery to bone marrow, which, in turn, increases iron absorption. Only a dozen cases of atransferrinemia with the complete absence of plasma transferrin have been reported [88].

Finally, oral iron ingestion does not cause iron overload except in patients with ineffective erythropoiesis and in genetically predisposed individuals [31]. African iron overload, which is seen in up to 10% of adults in rural societies in Sub-Saharan Africa, was thought to be exclusively caused by increased dietary iron through the consumption of a traditional beer containing high amounts of dissolved iron from its preparation in iron drums [89]. Recent studies of pedigrees, however, suggest that, in addition to a high dietary iron content, the disorder does, in fact, have a genetic component caused by variants in genes distinct from the *HFE* gene [90], although the putative has not yet been identified.

#### 3.2.2. Secondary Iron Overload

Secondary iron overload occurs in individuals who absorb or store excessive amounts of iron as a result of underlying diseases other than those previously mentioned or due to iatrogenic iron overload through frequent RBC transfusions or parenteral iron administration. Owing to the liver’s major role in iron homeostasis, hyperferritinemia in liver diseases may also be associated with an iron overload. Reduced liver function in chronic liver diseases is associated with impaired hepcidin and transferrin synthesis, hepatic iron overload (HIO), as well as increased levels of circulating NTBI [50,91]. Consequently, iron overload has been proposed as a cofactor in these conditions, but its exact role remains unclear [92]. While HH may cause severe iron overload if left untreated, secondary iron overload due to chronic liver diseases usually remains minimal to modest [93].

HIO is present in up to 50% of patients with alcoholic liver disease [50]. Cirrhosis due to chronic hepatitis C contributes to hepatic iron accumulation [50], and *HFE* variants such as C282Y heterozygosity have shown to accelerate hepatic fibrosis and HIO in these patients [94]. Porphyria cutanea tarda should be considered when hyperferritinemia is seen with a photosensitive rash and liver dysfunction. This is an acquired liver disease in 80% of clinical cases, and exogenous factors (e.g., alcohol, smoking, and hepatitis C) are a prerequisite for inducing HIO in these patients [1,48,95].

One third of patients with NAFLD and metabolic syndrome have increased body iron stores, known as dysmetabolic iron overload syndrome—a much more common condition than clinically recognized by physicians. These patients most often exhibit a normal transferrin saturation and have been found to have increased hepcidin concentrations with a mixed pattern of iron retention in both hepatocytes and macrophages [96]. Studies suggest this to be the result of hepcidin-resistance and a compensatory mechanism to prevent and counteract iron accumulation [96,97,98].

Although iron deficiency anemia affects nearly all chronic kidney disease and long term hemodialysis patients as the disease progress [99], iatrogenic iron overload attributed to RBCs transfusions due to hypoproliferative erythroid marrow and intravenous iron to ensure sufficient available iron during therapy is also recognized as a complication in some of these patients. Mild to severe HIO measured by MRI was observed in 84% of hemodialysis patients treated with erythropoietin and regular intravenous iron supplementation, in keeping with guidelines, although transferrin saturation was normal for all [100]. Hyperferritinemia seen in chronic kidney disease patients on hemodialysis is also partly a result of systemic inflammation, with ferritin levels correlating positively with the severity of it [101].

Prolonged parenteral administration of iron or the transfusion of RBCs in patients with chronic anemia such as thalassemias or dyserythropoietic, aplastic, sideroblastic, and hemolytic anemias will also most often result in iron overload. Tissue deposition becomes significant when more than 40 units of RBCs are transfused [23]. Iron is deposited in the macrophages of the RES prior to the iron-loading of the liver and heart parenchyma, ultimately leading to progressive heart and liver failure if left untreated [91].

Even in the absence of transfusions, a variety of blood cell disorders such as myelodysplastic syndrome, thalassemias, and other iron-loading anemias are associated with increased iron absorption. Hepcidin is down-regulated by signaling molecules associated with anemia and hypoxia upon these conditions [30]. Ineffective erythropoiesis, therefore, leads to low hepcidin levels, and, subsequently, increased intestinal iron absorption. The hormone erythroferrone, secreted by erythroid precursors, suppresses hepcidin and increases the amount of iron available for hemoglobin synthesis [102]. Loss of this hormone in thalassemic knock out mice led to the full restoration of hepcidin mRNA expression and a significant reduction in the iron content of the liver and spleen [103], making erythroferrone a potential therapeutic target upon secondary iron overload in anemias and other blood cell disorders.

## 4. Epidemiology

The prevalence of underlying conditions related to hyperferritinemia in various study settings are summed up in Table 2. Approximately 20% of Caucasian men have ferritin levels >300 μg/L, irrespective of age. In women, there is, however, a significant age distribution of ferritin due to menstruation and pregnancies. Among women aged 30–50 years, 3% have ferritin levels >200 μg/L, while corresponding levels are found in up to 17% of women at ages >70 years [104]. Ferritin elevations are often encountered in healthy individuals of Asian and African American descent, but presence of iron overload or C282Y homozygosity is rare in these groups. The biological basis and clinical significance of these ethnical differences is incompletely understood, but should be taken into account when interpreting an unexpected finding of hyperferritinemia [29].

Prevalence of hyperferritinemia in primary care has, on the whole, been determined by screening studies aimed at identifying patients with HH and iron overload. Only a minority of individuals with hyperferritinemia in such studies are *HFE* C282Y homozygotes [1,32]. For instance, among almost 100,000 multi-ethnic volunteers, 364 subjects had ferritin >1000 μg/L, but only 29 of those were C282Y homozygotes [32]. Although prevalence of ferritin >1000 μg/L appears to be more common in primary care patients than in such volunteer studies, only 180 patients had such levels among 1745 patients with ferritin >200 μg/L and transferrin saturation >30%. Out of those 180 subjects, only 29 were C282Y homozygotes in a high-prevalence area [9,72].

Annual incidence of ferritin >1000 μg/L in a secondary care setting was found to be 6.7% [105]. Incidence of extreme hyperferritnemia defined as ferritin >10,000 μg/L in a general hospital setting was, on the other hand, only 0.08% [106]. A study conducted in Japan on both out- and inpatients with ferritin >500 μg/L showed that 41% of subjects had multiple underlying conditions contributing to this increase. Furthermore, 70% of patients with ferritin levels >10,000 μg/L had multiple etiologies. The more underlying conditions a patient had, the higher their ferritin levels were [107]. Although the increase was not statistically significant, other studies have reported a similar association [44], suggesting that the rise in ferritin not only depends on a specific etiology, but is also progressively related to the number of underlying coexisting conditions.

## 5. Diagnostic Workup

In this overview, we have predominantly reviewed three major evidence-based practice guidelines on the diagnosis and management of HH. These include clinical guidelines developed by the British Society for Haematology (BSH), which also include contributions from gastroenterologists and hepatologists [80], the American Association for the Study of Liver Diseases (AASLD) [31], and the European Association for the Study of the Liver (EASL) [36]. Based on the lack of consistency between AASLD and EASL guidelines, with only 5.9% shared literature and approximately 50% of recommendations not overlapping, it has been suggested that the quality, consistency, and evidence base of current guidelines can be improved [108].

To better aid clinicians in the diagnostic workup of hyperferritinemia, we propose an algorithm for the identification of underlying causes and further management (Figure 5). In the majority of cases, a non-invasive diagnostic workup will identify the cause. Knowledge among clinicians of characteristic ferritin elevations in various conditions is of benefit, as patients presenting with an unexpectedly high ferritin for the initial diagnosis might serve as a clue for coexisting conditions and should prompt for further investigation. As mentioned earlier, genetic and environmental factors, age, and gender must also be taken into account when evaluating abnormal iron indices [26,32,104,109]. Table 2 is shown below.

Medical history must be thoroughly assessed in order to evaluate if there are any known clinical conditions that may cause a secondary increase in ferritin. Prolonged exposure to welding fumes has recently been suggested as a novel cause of systemic iron overload in a subgroup of welders [114], and occupational anamnesis may therefore be relevant. Family history of hyperferritinemia is also of interest, as this might indicate a genetic cause. If the patient has a known first-degree relative with HH, proceeding directly to *HFE* genetic testing should be considered [36].

It is often reasonable to repeat ferritin measurements over a period of time. When encountering a moderate and isolated ferritin <1000 μg/L in an otherwise healthy patient with normal transferrin saturation, a period of observation with lifestyle adjustments, if appropriate, and repeated measurements after two to three months may prove helpful in the diagnostic workup [1], although specific guidelines regarding this matter are lacking.

Longitudinal trends in ferritin establish the degree and rate of variations, which may assist the diagnostic workup. Alcohol-induced hyperferritinemia, hepatic steatosis, steatohepatitis, or any form of acute phase reactions usually cause fluctuating levels, contrary to HH or other genetic conditions characterized by hyperferritinemia such as HHCS which cause a relatively stable ferritin. Markedly elevated ferritin >10,000 μg/L should prompt the consideration of rare conditions such as adult-onset Still’s disease and HLH, although such levels may also be seen in more common conditions such as renal disease, liver disease, infection, or malignancy [44], as previously illustrated in Table 2.

If transferrin saturation is normal, secondary causes are of interest in the early phases of investigation. Relevant laboratory tests can usually be requested simultaneously when considering these, including hemoglobin, CRP, erythrocyte sedimentation rate (ESR), alanine aminotransferase, aspartate aminotransferase, gamma-glutamyl transferase, glucose, cholesterol, triglycerides, electrolytes, and creatinine [7,36]. Patients with a stable ferritin >1000 μg/L and/or abnormal liver function tests should be considered for referral and further investigation, regardless of transferrin saturation, as this is associated with an increased risk of hepatic fibrosis and is likely to be due to serious underlying pathology [7,80]. 

Finally, a slow and progressive increase in ferritin over time indicates an ongoing iron overload, and an elevated transferrin saturation can usually confirm this. Transferrin saturation >45% in women and >50% in men is generally regarded as elevated [31,36,80]. Upon such levels, iron-loading anemias must also be ruled out through the measurement of hemoglobin, reticulocytes, full blood count, and mean corpuscular volume as a minimum. It should be noted that acute infection, menses, and recent blood donation may reduce transferrin saturation to normal levels temporarily, despite the existence of HIO [115].

If an elevated transferrin saturation is confirmed, genetic testing for common *HFE* variants is recommended, even in asymptomatic patients [31,80]. Genetic testing of Caucasian patients presenting with ferritin >1000 μg/L is by some considered as relevant, as secondary causes are most often associated with a moderate increase in ferritin [7,80]. However, the additional finding of a raised transferrin saturation increases the likelihood of hyperferritinemia being related to *HFE* hemochromatosis, compared to that of an isolated ferritin >1000 μg/L in Caucasians. While transferrin saturation measurements are far less requested in clinical practice compared to ferritin, it is highly relevant when suspecting *HFE* hemochromatosis, as combined elevations in ferritin and transferrin saturation have proven to be an easy and inexpensive screening for appropriate candidates for *HFE* genetic testing [31,76,116,117].

Due to within-person biological variability and diurnal fluctuations of serum iron, some argue that, for the diagnosis of *HFE* hemochromatosis, the ideal specimen for iron studies is a fasting morning sample of transferrin saturation [36,118,119]. Several studies on fasting compared with random iron tests have, however, failed to confirm the added value of this practice, as it improved neither sensitivity nor specificity in detecting C282Y homozygotes [120,121,122]. Some guidelines therefore state that this is not necessary, given that borderline results are repeated or checked on a fasting morning sample if desired [1,7,31].

Non-*HFE* related iron overload may, in particular, present with elevated ferritin levels as the only and major indication of increased iron stores [119]. As mentioned earlier, neither ferritin nor transferrin saturation provide definite evidence for the presence or absence of HIO [123]. If the diagnostic workup is uncertain and iron overload cannot be ruled out, it is essential to evaluate iron burden. Clinical scenarios where the evaluation of iron burden should be considered are summed up in Table 3.

While liver biopsy remains the gold standard for iron quantification, non-invasive hepatic MRI has reduced the number of biopsies required and is the preferred method in many cases. When substantial HIO is detected through hepatic MRI, digenic inheritance or rare genetic causes of iron overload (Table 1) must be considered [1], and iron-loading anemias should be ruled out [31]. Finally, as iron depletion through phlebotomy remains the mainstay of treatment in HH, a trial of phlebotomy is an alternative invasive approach to determine the presence of an iron overload. When patients tolerate weekly phlebotomies without becoming anemic, this is usually proof of an iron overload being mobilized and removed from stores [7].

MRI is an adequate diagnostic method to detect severe iron overload. It is, however, less sensitive in detecting mild iron overload. Another disadvantage of MRI is that, in contrast to liver biopsies, it cannot detect patterns of iron overload and cellular distribution that might provide diagnostic clues [124]. Pure parenchymal iron overload is typically seen in *HFE* hemochromatosis and non-*HFE* HH, end stage cirrhosis, and iron-loading anemias. Mesenchymal or mixed iron overload is, on the other hand, mostly associated with NAFLD, alcoholic liver disease, dysmetabolic iron overload, ferroportin disease, porphyria cutanea tarda, and viral hepatitis [36].

While genetic analysis for detecting common *HFE* variants is widely available, this is not the case for other iron-loading genetic variants. In many cases of non-*HFE* and other miscellaneous iron-loading conditions where a genetic cause is suspected, the specific genes likely to be involved must be deduced from clinical features in the patient and through family medical history. Prior to genetic testing and in unclear cases, family members should also be evaluated for iron overload [31]. However, when MRI shows no signs of HIO despite a stable hyperferritinemia, rare abnormalities such as benign hyperferritinemia and HHCS might prove to be the underlying cause. Genetic testing is not an absolute necessity in such cases. For instance, the demonstration of hyperferritinemia and cataracts in at least two members of the same kinship is usually diagnostic for HHCS [7].

For other rare genetic causes related to hyperferritinemia, simple blood work might be sufficient for the diagnostic workup. Aceruloplasminemia most often presents with a low transferrin saturation in the setting of an atypical microcytic anemia with paradoxical hyperferritinemia. Prompt diagnosis and treatment of aceruloplasminemia is essential in order to prevent irreversible neurological complications. Serum ceruloplasmin, which is a widely available test, should therefore be requested in selected cases, with the absence of serum ceruloplasmin usually being diagnostic [125,126]. Although less available, measurements proving reduced β-glucocerebrosidase enzyme activity usually confirm the diagnosis of Gaucher disease. Enzyme replacement has shown to be highly successful for the management of systemic manifestations in these patients [127].

## 6. Management and Follow Up

Hyperferritinemia remains unexplained after a thorough evaluation in many patients, as observation and follow up might be a more appropriate practice than invasive investigation [7]. Patients with mildly increased ferritin levels (300–1000 μg/L) where iron overload is unlikely should be given lifestyle advice such as alcohol abstinence, improved glycemic control, weight reduction, and diet instruction to lower triglyceride concentrations when lifestyle behaviors have been found as a potential cause. Upon alcohol-induced hyperferritinemia, abstinence has shown to reduce ferritin significantly by 50% in 15 days, whilst return to normal levels may take more than six weeks [48]. Ferritin should subsequently be rechecked after initiating lifestyle adjustments for further management [33].

However, if an iron overload is detected, therapy in order to induce iron depletion in these patients should be considered, and general principles regarding the management of these patients are discussed below.

### 6.1. Primary Iron Overload

Out of the three guidelines reviewed, EASL was the only guideline that concluded on a case definition for *HFE* hemochromatosis: C282Y homozygosity with increased body iron stores with or without clinical symptoms [36]. Management related to other disease-associated *HFE* genotypes are, therefore, not discussed in their guidelines, whilst BSH and AASLD guidelines did not specify any genotypic feature for diagnosis.

Ferritin is a highly sensitive marker for iron overload in C282Y homozygotes, and normal levels essentially rule out the presence of an iron overload. Ferritin levels are distributed fairly uniformly among C282Y homozygotes, with over 70% of patients presenting with levels <1000 μg/L at the time of diagnosis [36,80,128]. Ferritin also robustly predicts the risk of cirrhosis [18], and levels >1000 μg/L showed 100% sensitivity and 70% specificity for the identification of cirrhosis in a study of asymptomatic C282Y homozygotes, whilst significant fibrosis can essentially be excluded when transaminases are normal with ferritin <1000 μg/L [1,129,130]. Accordingly, all three guidelines recommend a liver biopsy to stage liver disease in C282Y homozygotes with elevated liver enzymes and/or ferritin >1000 μg/L [31]. AASLD recommend the same strategy in C282Y/H63D compound heterozygotes, whilst further specification of genotypes lack in the BSH guidelines.

A study on alcohol consumption in C282Y homozygotes reported that >60% of those consuming >60 g alcohol/day had cirrhosis, compared to <7% in the group consuming less [131]. In accordance with these findings, it is not required to perform a liver biopsy and/or elastography when ferritin is <1000 μg/L in the absence of excess alcohol consumption and elevated serum liver enzymes [31]. On the contrary, abnormal iron indices are found in 50% of patients with liver disease such as alcoholic liver disease, NAFLD, and chronic viral hepatitis. AASLD recommends liver biopsy to be considered also in patients who are not C282Y homozygous, but in whom elevated serum iron indices and liver enzymes are detected [31].

Hepatocellular carcinoma accounts for approximately 30% of *HFE* hemochromatosis-related deaths. Patients with cirrhosis have a 100-fold increase in the risk of developing hepatocellular carcinoma and should continue to be screened every six months following phlebotomy, including hepatic ultrasound and alfa-fetoprotein monitoring [1,31,36,80]. Although randomized clinical trials are lacking, the long-term survival of *HFE* hemochromatosis patients is thought to be approximately the same as that of the general population, with significantly reduced morbidity and mortality compared to untreated patients when phlebotomy is initiated before development of cirrhosis and diabetes [80,132,133,134]. Compliance is high; however, phlebotomy is not suitable for all patients, due to religious beliefs, comorbidities, or physiological or psychological intolerance [29].

Ferritin has been shown to remain relatively stable in some C282Y homozygotes over a 12-year period, and the majority of those who are likely to develop ferritin levels >1000 μg/L do so by the mean age of 55 years [117]. No studies on the phlebotomy of HH patients provide data on the optimal initiation timepoints, optimal frequency, or optimal endpoints, and current guidelines are thus empirical. Reliable indicators as to who will develop complications in HH are also unavailable to date, and in the absence of results from controlled trials, prophylactic phlebotomy in all fit patients with a biochemical iron overload with or without clinical features is generally favorable.

A study on primary care male C282Y homozygotes predicted that 40% of those presenting with ferritin levels of 300–900 μg/L have increased iron stores >4 g [36,135]. Male *HFE* hemochromatosis patients with ferritin >300 μg/L and transferrin saturation >50%, and female patients with ferritin >200 μg/L and transferrin saturation >45%, should undergo weekly or biweekly phlebotomy, removing a blood volume of 400–500 mL, equivalent to 200–250 mg of iron, each session. Patients with normal iron indices may be suitable for blood donation and having ferritin and transferrin saturation annually monitored [80]. Recommendations regarding the evaluation of treatment response and tolerance during therapeutic phlebotomy are also empirical, but hemoglobin should generally be monitored at each session to guide frequency and volume, and in order to evaluate tolerance [31,36,80].

Transferrin saturation usually remains elevated until iron stores are depleted during therapeutic phlebotomy, whereas ferritin may initially fluctuate and begin to fall progressively with iron mobilization [36]. Accordingly, ferritin measurements are usually sufficient for monitoring iron depletion. The appropriate frequency of ferritin measurements during the initial phase is also largely based on empirical observations, and recommendations range from every three months, upon high concentrations and more frequently as levels reach a normal range [36], to every 10–12 phlebotomies [31] or monthly [80]. Furthermore, recommendations regarding target ferritin and the endpoint of therapeutic phlebotomy range from 20–30 μg/L to 50–100 μg/L. In patients with non-*HFE* HH where HIO is detected, phlebotomy should be initiated to the same endpoints as discussed above [31].

Patients with HH need lifelong follow-up, and subjects who have had an iron overload should never stop having their iron status monitored [36]. Frequency of maintenance phlebotomy varies due to variable rates of iron accumulation [31]. Not all C282Y homozygotes show a progressive increase in ferritin, presumably as a result of a steady state developing when a certain level of iron is reached. Additionally, many patients appear to lose iron-loading tendency over time, with a decreasing need for phlebotomy [80,136], and iron absorption has been shown to decline in patients as their iron load increases, suggesting a physiologically self-limited iron accumulation [137].

Recommendations regarding surveillance after therapeutic phlebotomy differ, as BSH advise ferritin to be kept <50 μg/L and transferrin saturation <50%, whilst AASLD and EASL recommend ferritin to be kept between 50–100 μg/L and transferrin saturation to not be monitored before considering maintenance phlebotomy. There is uncertainty about the appropriate management of patients with increased transferrin saturation, despite ferritin <50 μg/L. None of the three guidelines comment on how often ferritin should be monitored in the maintenance phase, but ferritin measurements at six months and one year post therapeutic phlebotomy are valuable in assessing the need for maintenance therapy [29]. Supplemental vitamin C accelerates the mobilization of iron to a level that may saturate circulating transferrin, potentially resulting in an increase of free radical activity—therefore, this ought to be avoided in iron-loaded patients, particularly those undergoing phlebotomy [31,36].

Recommendations regarding family screening also vary among guidelines. BSH and AASLD recommended all first-degree relatives of C282Y homozygotes to be offered screening, while EASL recommend screening to be restricted to siblings only. Based on a cost-effective analysis, EASL also suggests unaffected spouses be screened in order to establish the need for testing children later in life [138]. While BSH recommends such family screening be restricted to C282Y homozygotes, EASL and AASLD recommend extended family screening for all *HFE* hemochromatosis patients, without specifying proband genotype [80].

Testing for both genotype and phenotype simultaneously at a single visit is recommended in order to detect early disease and to prevent complications upon family screening. Therapeutic phlebotomy can be initiated if ferritin is increased in adult relatives who are found to be C282Y homozygous or C282Y/H63D compound heterozygous [31]. Otherwise, if ferritin is normal, a yearly follow-up with iron studies is sufficient [31]. These individuals may also be encouraged to volunteer as blood donors [1].

### 6.2. Secondary Iron Overload

Phlebotomy is also useful in certain forms of secondary iron overload and is clearly indicated in patients with porphyria cutanea tarda towards ferritin <25 μg/L, in order to reduce skin manifestations [139,140]. Currently, phlebotomy is not recommended in mild secondary iron overload due to chronic hepatitis C, although it has been shown to reduce transaminase levels with a marginal improvement in histopathology [141].

Excess body iron appears to be associated with adverse outcomes, insulin resistance, and accelerated disease progression in NAFLD, and iron depletion upon phlebotomy improved insulin sensitivity in these patients [142,143]. A reduction in ferritin, which may signify a beneficial effect on liver fibrosis or the long-term outcome of liver disease, is also observed, and phlebotomy generally appears to be safe in these patients until ferritin levels reach 50–100 μg/L [144]. Nevertheless, results are controversial, and randomized trials of phlebotomy as treatment for NAFLD failed to show improvement in prognostic markers. It has therefore been argued that the role of phlebotomy in patients with increased ferritin levels associated with liver disease other than HH is of little benefit [1,145]. There is no evidence of phlebotomy being of benefit in alcoholic liver disease [31].

Although frequent measurements of ferritin is recommended in order to estimate iron burden in secondary iron overload as a result of transfusions, iron burden can also be easily calculated [124]. Patients with secondary iron overload due to ineffective erythropoiesis or frequent transfusions should be considered for iron chelation, as phlebotomy, for obvious reasons, is not indicated in these patients [146]. Only a small percentage of body iron is accessible, and chelator molecules should ideally be continuously present.

Deferoxamine has been the standard iron chelator of use for three decades and is usually administered subcutaneously. In thalassemia major, deferoxamine has shown to improve survival and reduce the prevalence of major complications related to iron overload [147,148]. A recently published randomized clinical trial showed longer event-free survival in lower-risk myelodysplastic syndrome iron-overloaded patients receiving iron chelation, compared to those who did not [149]. Monitoring treatment response upon chelation therapy becomes difficult, as ferritin can be misleading in secondary iron overload, and repeated liver biopsies may be required. Whilst this procedure could potentially be complicated by hemorrhage, MRI has shown promise in providing a non-invasive method to evaluate iron overload in these patients also [31,150].

Finally, studies on deferoxamine and its potential use in HH patients unable to undergo phlebotomy are ongoing [151], and deferoxamine has proven to be just as effective as phlebotomy at inducing iron depletion in C282Y homozygotes. Compliance is, however, a major problem upon chelation treatment, and for this reason, deferoxamine would generally not be a satisfactory iron depletion strategy for all HH patients [136,152,153].

## 7. Conclusions and Future Perspective

It is becoming clear that extracellular ferritin subsumes many functions unrelated to its classic role as an intracellular iron storage protein. Fundamental aspects of the biology of ferritin are still unknown and are topics of active debate that require further experiments.

Underlying causes of hyperferritinemia can most often be determined through simple clinical assessment and additional laboratory tests. However, mixed etiologies are often present, and elevated ferritin levels are not always pathological, but rather seen as a result of benign genetic conditions or biological variations.

In clinical medicine, elevated ferritin levels have, for a long time, been recognized as a non-specific finding most often seen as secondary to acute or chronic inflammation, chronic alcohol consumption, liver disease, renal failure, and metabolic syndrome. When transferrin saturation is normal, these conditions should be the priority for investigation, whilst common *HFE* variants and iron-loading anemias should be ruled out when transferrin saturation is elevated. When the underlying cause of hyperferritinemia still remains unknown after common causes and iron overload have been ruled out, annual follow-up might seem more appropriate than further invasive diagnostic workup.

There is still a lack of understanding of why clinical presentation dramatically varies in primary iron overload. Studies to identify possible genetic bias in transferrin saturation and ferritin are needed. There is still more to discover in primary iron overloading disorders, and variants of the genes currently associated with HH probably do not account for all clinical cases.

With regard to the expression of hepcidin to a variety of stimuli, it is possible that this may serve as a therapeutic target in iron-loading conditions and as an adjunct treatment to phlebotomy in the induction phases of therapeutic phlebotomy. The deliberate manipulation of iron loss through the gut epithelium or kidneys are two additional possible approaches, and chelation therapy has shown promising results in HH patients who cannot undergo phlebotomy therapy. Iron-depleting strategies may also become a treatment for altering innate immunity and to enhance host defense against invading pathogens and tumor progression in the future.

## Figures and Tables

**Figure 1 jcm-10-02008-f001:**
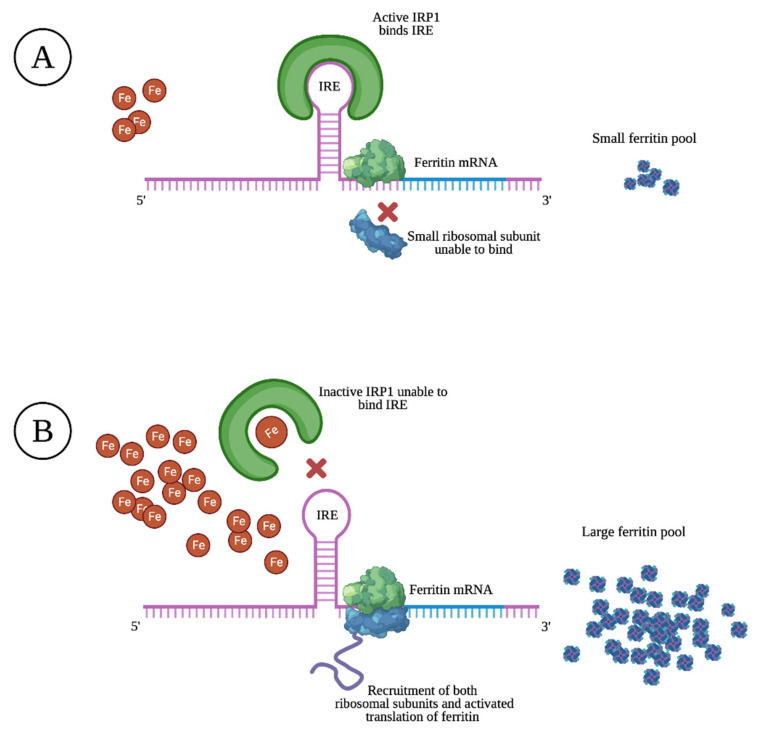
Schematic overview of regulated ferritin translation by the iron responsive element (IRE)/iron regulatory protein 1 (IRP1) system. Upon low intracellular iron levels (**A**), IRP1 binds to IRE, inhibiting the recruitment of the small ribosomal unit to mRNA, which results in a small ferritin pool. Upon high intracellular iron levels (**B**), iron binds to IRP1, causing a transformational change that dissociates IRP1 from IRE. Both ribosomal subunits are now recruited to the mRNA and the translation of ferritin is activated. This coordinated IRP1/IRE binding with respect to iron levels stabilizes cellular iron through the synthesis of ferritin.

**Figure 2 jcm-10-02008-f002:**
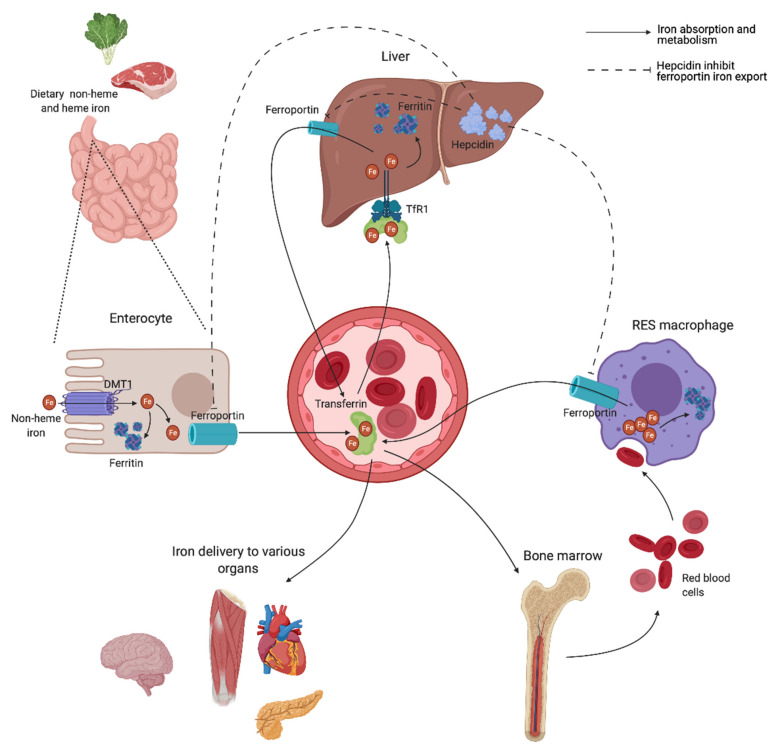
Body iron homeostasis and the hepcidin–ferroportin axis. Dietary non-heme is absorbed at the apical site of enterocytes of the jejunum through divalent metal transporter 1 (DMT1). One portion remains stored as ferritin inside the enterocyte, while the rest is transferred through the basolateral site via ferroportin. Iron subsequently binds transferrin and is further distributed throughout the body. Most iron is distributed to bone marrow for hemoglobin production. Senescent erythrocytes are phagocytosed by macrophages of the reticuloendothelial system (RES), and iron is catabolized from hemoglobin before subsequently re-entering circulation. Hepcidin produced in the liver regulates the systemic iron balance through binding to ferroporrtin. This binding facilitates the lysosomal degradation of the iron exporter, ultimately resulting in decreased serum iron concentrations.

**Figure 3 jcm-10-02008-f003:**
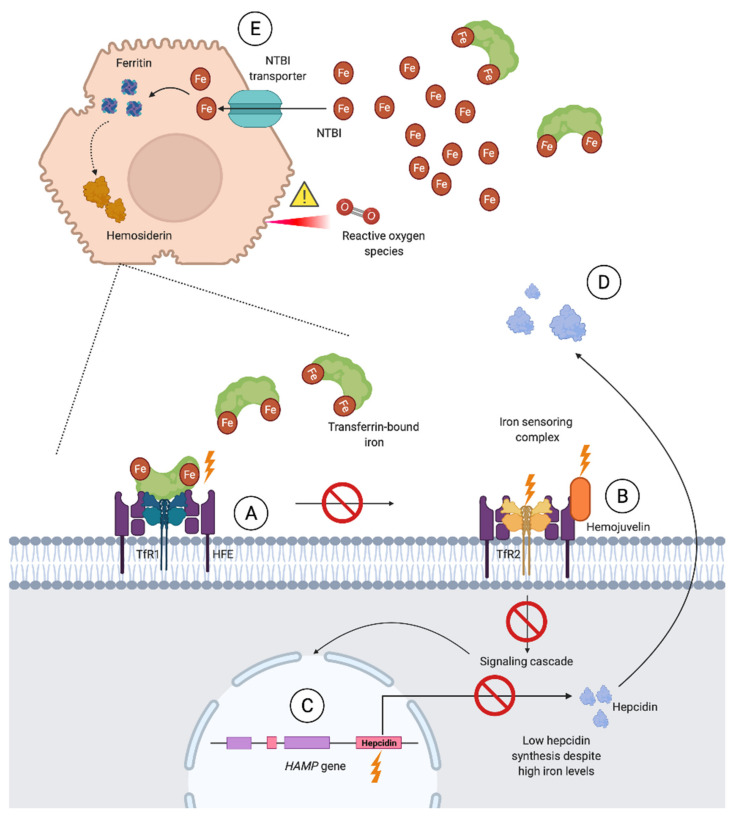
Schematic overview of hereditary hemochromatosis (HH) associated with impaired hepcidin expression. *HFE* disease-associated variants (type 1 HH) cause impaired assembly of the iron sensing complex (**A**), as transferrin binding to transferrin receptor 1 (TfR1) normally competes with and releases HFE to interact with transferrin receptor 2 (TfR2). Disease-associated variants of the *HJV* (type 2A HH) and *TFR2* (type 3 HH) genes, encoding subunits of the iron sensing complex, cause impaired function of this complex which normally regulates hepcidin transcription through a signaling cascade (**B**). Disease-associated variants of the *HAMP* (hepcidin) gene (type 2B HH) cause reduced hepcidin levels, despite a functioning iron sensing complex (**C**). An impaired hepcidin–ferroportin axis (**D**) results in uncontrolled intestinal iron absorption and the release of iron stores from macrophages and parenchymal cells, and, accordingly, high serum iron levels. When transferrin saturation is highly elevated, the transferrin iron-binding capacity is exceeded and non-transferrin bound iron (NTBI) enters circulation. NTBI is a more toxic form than transferrin-bound iron, capable of producing reactive oxygen species, which are involved in the cellular damage seen upon iron-loading conditions, for example, liver fibrosis and cirrhosis. NTBI is eliminated from circulation through transporters mainly expressed on hepatocytes (**E**). Here, it is subsequently stored in iron-storing complexes such as ferritin and hemosiderin to limit cellular damage.

**Figure 4 jcm-10-02008-f004:**
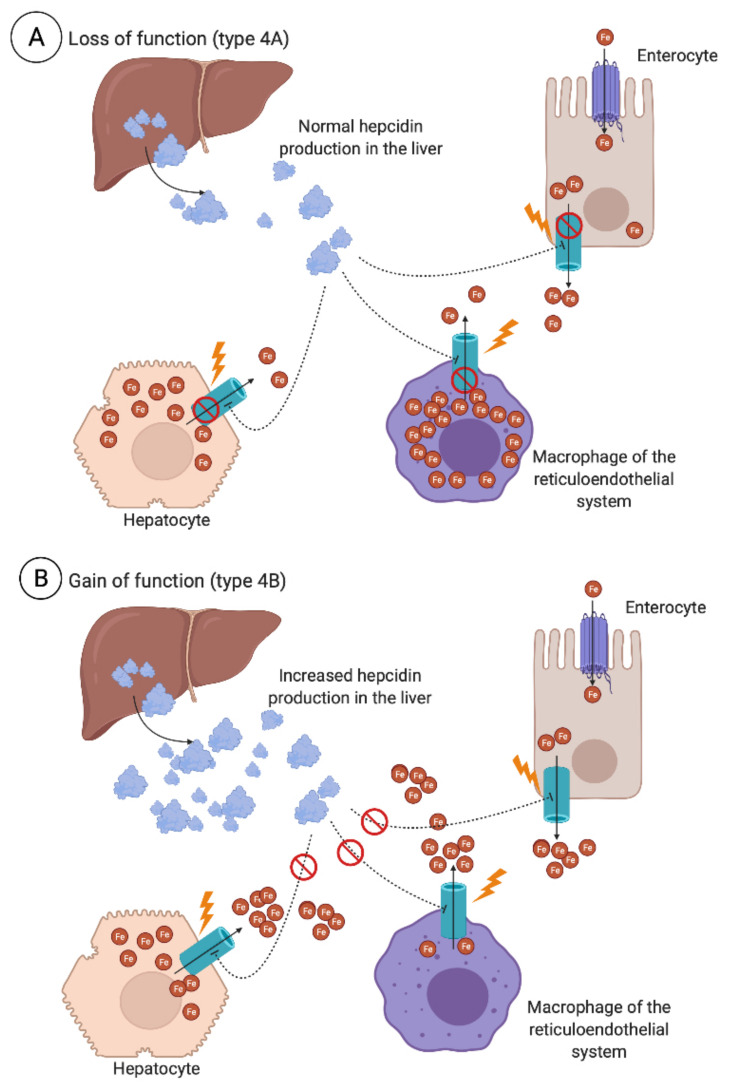
Variants of the *FPN1* (SLC40A1) gene encoding the iron exporter ferroportin involved in type 4 hereditary hemochromatosis (HH). Loss-of-function variants (**A**) impair the iron-export capability or expression of ferroportin, leading to iron accumulation mainly in cells such as tissue macrophages and discrete accumulation in parenchymal cells, decreased iron delivery to circulating transferrin causing an inappropriately low transferrin saturation, and decreased iron delivery for hemoglobin production. This is referred to as ferroportin disease, or type 4A HH. Gain-of-function variants (**B**), however, make ferroportin resistant to hepcidin-induced degradation, which normally inhibits the export of iron. This results in a similar impairment of the hepcidin–ferroportin axis, as seen in type 1-3 HH, with high serum iron concentrations and transferrin saturation, and iron mainly accumulating in parenchymal cells and hepatocytes. This is often referred to as type 4B HH.

**Figure 5 jcm-10-02008-f005:**
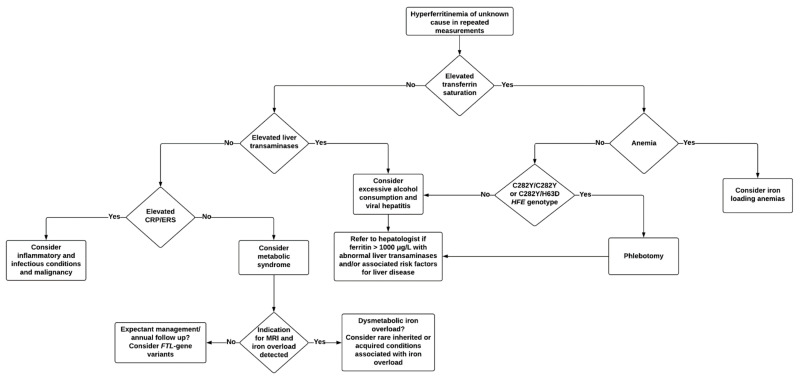
Diagnostic workup and management in hyperferritinemia of unknown cause. CRP: C-reactive protein; ESR: erythrocyte sedimentation rate; MRI, magnetic resonance imaging.

**Table 1 jcm-10-02008-t001:** Underlying conditions in hyperferritinemia with and without an associated iron overload.

Hyperferritinemia without iron overload	**Common causes**
	Cellular damage
	Metabolic syndrome and obesity
	Insulin resistance/diabetes mellitus
	Excessive alcohol consumption
	Inflammatory and infectious conditions (septic shock, COVID-19)
	Malignancy (solid and hematological)
	**Rare causes**
	Benign hyperferritinemia/HHCS
	Immune-mediated syndromes (primary and secondary HLH, adult-onset Still’s disease) Gaucher disease
Hyperferritinemia with or without iron overload	**Common causes**
	Chronic liver disease (cirrhosis, alcoholic liver disease, NAFLD, viral hepatitis, porphyria cutanea tarda)
Hyperferritinemia with iron overload	**Common causes**
	*HFE* hemochromatosis
	Dysmetabolic iron overloading syndrome
	Iron-loading anemias (congenital or acquired)
	Iatrogenic iron overload (RBC transfusion, parenteral iron administration)
	African iron overload
	**Rare causes**
	Non-*HFE* hereditary hemochromatosis Ferroportin disease
	Aceruloplasminemia/hypoceruloplasminemia Atransferrinemia/hypotransferrinemia

COVID-19, coronavirus disease 2019; HHCS, hereditary hyperferritinemia cataract syndrome; HLH, hemophagocytic lymphohistiocytosis; NAFLD, non-alcoholic fatty liver disease; RBC, red blood cell.

**Table 2 jcm-10-02008-t002:** Prevalence of common causes associated with various degrees of hyperferritinemia in adult and pediatric study populations.

Ferritin Inclusion Criteria (μg/L)	>500	>500	≥1000	>1000	≥1500	>2000	≥10,000	>10,000	>10,000	>10,000	>50,000
**Recruitment method**	Adult outpatients and inpatients of a general hospital in Japan	Pediatric outpatients and inpatients of a tertiary care population in America	Adult general hospital population in America	Adult white patients recruited through screening study in America	Adult tertiary care population in England	Adult inpatients of a general hospital in France	Unspecified general hospital population in England	Adult and pediatric tertiary care population in America	Adult outpatients and inpatients of a general hospital in Japan	Adult and pediatric tertiary care population in America	Adult tertiary care population in America
**Reference**	[107]	[43]	[105]	[110]	[34]	[111]	[106]	[112]	[107]	[113]	[44]
**Number of patients included**	1394	330	95	59	150	77	23	86 †	111	628 ¥	113
**At least two etiologies identified ¶ (%)**	41	N/A	16.8	N/A	46	N/A	N/A	N/A	70	N/A	86.7
Unspecified infection (%)	44.8	18.5	15.8	No data	No data	18.2	4.3	8.1	47.3	13.9	46
HIV infection (%)	0.8	No data	16.8	No data	0.7	0	8.7	No data	2.7	No data	0
Malignancy (%)											
* Hematological*	12.0	No data	No data	No data	No data	26	8.7	16.3	30.9	25.1	32
* Solid*	26.3	No data	No data	No data	No data	0	13.0	2.3	20.9	2.5	4
* Unspecified*	0	7	17.9	15	19.3 §	0	4.3	0	0	No data	No data
Rheumatological/inflammatory disease (%)	6.3	8.2	No data	No data	32.6 ‡	1.3	4.3	No data	14.5	1.8	18
Hepatocellular injury (%)	20.3	2.7	20	12	35.3	6.5	22	27	35.5	15.1	54
Renal failure (%)	20.2	No data	17.9	No data	28	16.9	No data	No data	23.6	9.4	65
Iron overload (%)											
* Primary*	No data	No data	1.1	41	8.6	No data	N/A*	0	No data	0.5	No data
* Secondary*	No data	25.2	10.5	No data	17.3	No data	N/A*	35	No data	4.8	No data
* Unspecified*	6.7	No data	No data	No data	No data	16.8	N/A*	0	21.8	No data	12
HLH (%)	1.4	3	No data	No data	No data	14.3	0	9.3	13.6	16.7	20 ¢
Hematological disorders (%)	0.2	4.8	10.5	2	25.3	0	8.7	1	1.8	8.3	4
Other causes (%)	10.8	27.6	7.3	22 ¡	11.3	0	13	0	3.6	1.6	2
Unknown (%)	0	3	0	8	2	0	13	1	0	0.3	0

Note: This table is based on a selection of 10 studies on the underlying causes of hyperferritinemia in various study populations. Criteria and case definitions therefore vary between studies. HIV: human immunodeficiency virus; HLH: hemophagocytic lymphohistiocytosis. No data are reported when the exclusion of a category cannot be performed due to lacking information. N/A, not applicable, studies only reported a single cause for each subject. N/A*, not applicable, study excluded hemochromatosis patients. ¶, prevalence of study participants where multiple etiologies were identified. ¡, including 17% excessive alcohol consumption and 5% elevated liver enzymes. §, reported as neoplasia through histologically proven neoplastic disease, without further differentiation of malign or benign growth. ‡, including 14% autoimmune disease and 18.6% inflammatory disease, the latter defined as raised C-reactive protein and/or erythrocyte sedimentation rate on more than one consecutive test, and recognized active inflammatory disease, without further differentiation of infectious cause or not. †, study population included 22% children. ¥, study population included 7.2% children. ¢, including 3% macrophage activation syndrome.

**Table 3 jcm-10-02008-t003:** Cases where the assessment of hepatic iron burden through hepatic magnetic resonance imaging or liver biopsy should be considered, according to guidelines and recommendations. A trial of phlebotomy as an alternative approach to estimate iron burden may also be appropriate.

Clinical Cases where the Assessment of Hepatic Iron Burden Should Be Considered
Transferrin saturation >50% without any clear cause
Ferritin >1000 μg/L with a normal transferrin saturation and no obvious explanation
Ferritin >1000 μg/L with liver disease risk factors (e.g., alcohol, viral hepatitis, obesity)
C282Y homozygotes with ferritin >1000 μg/L and/or raised liver transaminases
Hyperferritinemia with a history of multiple transfusions
Hyperferritinemia and multiple conditions associated with elevated ferritin identified
Ferritin increasing over time despite a suspected underlying cause being considered controlled

## Data Availability

Not applicable.

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
