# Peer review of "Hyperferritinemia—A Clinical Overview"

_jcm, 2021, doi:10.3390/jcm10092008_

Round 1
Reviewer 1 Report
Notwithstanding the vastity of the topic, the paper is well-organized and pleasant to read. Adding some insights from recent papers that I've suggested below in the comments can improve the quality of the manuscript. My comments/suggestions are the following:
Line 12: low SF is highly specific for iron deficiency (not necessarily iron deficiency anemia)
Lines 28-29: "... , low SF ... of iron deficiency" (again, not necessarily iron deficiency anemia)
Line 95: ...requires hephaestin or ceruloplasmin
Table 1:
- HF without IO: septic shock and COVID-19 are not so rare conditions associated to HF, I would mention them among "inflammatory and infectious conditions".
I would mention, as rare causes, inherited metabolic disorders, such as Gaucher disease (Marchi G et al, AJH 2020 https://doi.org/10.1002/ajh.25752).
- HF with IO: I would not mention CKD and long term haemodialysis as a cause of IO. I'm aware of the study by Rostoker in Am J Med 2012 cited by the Authors, but in clinical practice CKD and long term haemodialysis are more commonly associated to IDA of clinical significance instead of IO, even if many patients have a spurious HF, as the Authors state later in the text.
I would mention Ferroportin disease, which has been recently classified apart from non HFE hemochromatosis (Pietrangelo A, Haematologica 2017 doi: 10.3324/haematol.2017.170720), and prolonged exposure to welding fumes (Mariani R et al, Liv Int 2021 https://doi.org/10.1111/liv.14874) as rare causes of HF with IO.
Lines 274-276: The H63D variant has been recently reconsidered, also when associated to C282Y. If a subject with C282Y/H63D compound heterozygosity shows IO, he/she should be investigated for additional acquired or genetic causes of IO as the Authors state later in the text. Therefore, I would reformulate sentence at line 275 using "even when" instead of "unless".
Lines 320-327: For Ferroportin disease I would mention also some clinical features which are different from classical Hemochromatosis phenotype, such as possible mild anemia, especially after phlebotomies, which are less tolerated, normal/low TSAT, and IO prevalent in spleen macrophages rather than in liver hepatocytes.
Line 329-331: It could be reported that aceruloplasminemia has the peculiarity to show brain and systemic iron accumulation.
Lines 390-391: I would mention also dyserythropoietic and aplastic anemia.
In the diagnostic work up I suggest to add:
- professional anamnesis, since prolonged exposure to welding fumes has been recently associated to systemic IO (Mariani R et al, Liv Int 2021 https://doi.org/10.1111/liv.14874)
- ceruloplasmin dosage in selected cases, since it is widely available and it is crucial for an early diagnosis of aceruloplasminemia.
- beta-glucocerebrosidase activity in selected cases to detect Gaucher disease (Marchi G et al, AJH 2020 https://doi.org/10.1002/ajh.25752), an overlooked and potentially treatable disorder.
- a brief diagnostic work up of iron loading anemias, for example assessment of reticulocytes, hemolysis indexes, Hb electrophoresis, etc, eventually citing bibliography for further insights (Brissot P et al Mutat Res 2018; Donker A et al Blood 2014).
Lines 456-457: I suggest the Authors to stress the fact that TSAT is much less requested than ferritin in clinical practice, but it is much more informative in suspecting HFE Hemochromatosis. My opinion, and the one of other experts in iron disorders, is that genetic test should be performed in Caucasian patients with high TSAT rather than in those with SF >1000 ug/l.
Author Response
Notwithstanding the vastity of the topic, the paper is well-organized and pleasant to read.
We are grateful for the mainly positive comments regarding our present manuscript, in addition we thank for the comments, which indeed have helped us to improve our current paper.
Adding some insights from recent papers that I've suggested below in the comments can improve the quality of the manuscript. My comments/suggestions are the following:
Line 12: low SF is highly specific for iron deficiency (not necessarily iron deficiency anemia)
This is corrected.
Lines 28-29: "... , low SF ... of iron deficiency" (again, not necessarily iron deficiency anemia)
This is corrected.
Line 95: ...requires hephaestin or ceruloplasmin
Hephaestin and ceruloplasmin are now mentioned in the text (Lines 98-100).
Table 1:
- HF without IO: septic shock and COVID-19 are not so rare conditions associated to HF, I would mention them among "inflammatory and infectious conditions".
We have mentioned septic shock and COVID-19 under common causes (inflammatory/infectious conditions) in Table 1.
I would mention, as rare causes, inherited metabolic disorders, such as Gaucher disease (Marchi G et al, AJH 2020 https://doi.org/10.1002/ajh.25752).
We have implemented Gaucher disease as a rare cause of hyperferritinemia without iron overload in Table 1 and in the text (Lines 258-262)
- HF with IO: I would not mention CKD and long term haemodialysis as a cause of IO. I'm aware of the study by Rostoker in Am J Med 2012 cited by the Authors, but in clinical practice CKD and long term haemodialysis are more commonly associated to IDA of clinical significance instead of IO, even if many patients have a spurious HF, as the Authors state later in the text.
We have removed CKD and long term haemodialysis as a cause of IO in Table 1 and mentioned IDA as the major complication in these patients (Babitt JL et al, JASN 2012 https://doi.org/10.1681/ASN.2011111078 (Lines 418-427), while still mentioning iatrogenic iron overload as a complication in some of these patients.
I would mention Ferroportin disease, which has been recently classified apart from non HFE hemochromatosis (Pietrangelo A, Haematologica 2017 doi: 10.3324/haematol.2017.170720), and prolonged exposure to welding fumes (Mariani R et al, Liv Int 2021 https://doi.org/10.1111/liv.14874) as rare causes of HF with IO.
We have added Ferroportin disease as a separate rare cause of HF with IO, and briefly discussed potential occupational exposure as a risk of iatrogenic iron overload.
Lines 274-276: The H63D variant has been recently reconsidered, also when associated to C282Y. If a subject with C282Y/H63D compound heterozygosity shows IO, he/she should be investigated for additional acquired or genetic causes of IO as the Authors state later in the text. Therefore, I would reformulate sentence at line 275 using "even when" instead of "unless".
This is corrected.
Lines 320-327: For Ferroportin disease I would mention also some clinical features which are different from classical Hemochromatosis phenotype, such as possible mild anemia, especially after phlebotomies, which are less tolerated, normal/low TSAT, and IO prevalent in spleen macrophages rather than in liver hepatocytes.
Clinical features and non-aggressive phlebotomy regimen as treatment in Ferroportin disease have been implemented in the text (Lines 346-355).
Line 329-331: It could be reported that aceruloplasminemia has the peculiarity to show brain and systemic iron accumulation.
Brain iron accumulation and neurological symptoms in aceruloplasminemia are now mentioned (Lines 364-366).
Lines 390-391: I would mention also dyserythropoietic and aplastic anemia.
This has been added (Line 429).
In the diagnostic work up I suggest to add:
- professional anamnesis, since prolonged exposure to welding fumes has been recently associated to systemic IO (Mariani R et al, Liv Int 2021 https://doi.org/10.1111/liv.14874)
We have added the above mentioned study and occupational anamnesis in the diagnostic workup (Lines 494-497).
- ceruloplasmin dosage in selected cases, since it is widely available and it is crucial for an early diagnosis of aceruloplasminemia.
This has been added in the text (Lines 591-597).
- beta-glucocerebrosidase activity in selected cases to detect Gaucher disease (Marchi G et al, AJH 2020 https://doi.org/10.1002/ajh.25752), an overlooked and potentially treatable disorder.
This has been added in the text (Lines 597-600)
- a brief diagnostic work up of iron loading anemias, for example assessment of reticulocytes, hemolysis indexes, Hb electrophoresis, etc, eventually citing bibliography for further insights (Brissot P et al Mutat Res 2018; Donker A et al Blood 2014).
We have suggested a simple panel of appropriate laboratory tests that we regard as a minimum for this purpose (Line 529-531).
Lines 456-457: I suggest the Authors to stress the fact that TSAT is much less requested than ferritin in clinical practice, but it is much more informative in suspecting HFE Hemochromatosis. My opinion, and the one of other experts in iron disorders, is that genetic test should be performed in Caucasian patients with high TSAT rather than in those with SF >1000 ug/l.
We have implemented this key point in the text (Lines 535-544).
Reviewer 2 Report
This is a comprehensive and clearly written non-systematic review of potential causes and effects of hyperferritinemia, including guidelines for diagnostics and follow-up.
-In the abstract and 1st paragraph, 'low SF is highly specific for iron deficiency anaemia': this should just be 'iron deficiency' as you are not referring to any hemoglobin measurement,
-In Figure 2, it is not clear what the lines and arrows represent; please add text.
-Table 1: what is 'ethnicity associated hyperferritinemia'? Please be more specific.
-Page 6/7: 'world pandemic' is a pleonasm. It is then stated that 'all patients with COVID-19 should be screened for hyperinflammation', which does not make sense for mildly symptomatic cases.
-Figure 5: please add definitions of hyperferritinemia, anemia (which is misspelled as anaemia here) etc. For consistency: instead of 'rule out...' in some of the boxes, delete those words and and add yes/no to the arrows coming from those boxes.
-Table 3: how does this relate to Figure 5?
-Page 19: 'Patients with mildly increased SF (300-1000 μg/L) where iron overload is unlikely and other causes are not evident, should be given lifestyle advice such as alcohol abstinence, improved glycaemic control, weight reduction and diet instruction to lower triglyceride concentrations.' This does not make sense, unless these lifestyle behaviors have been found as a potential cause.
-Abbreviations: figures should be self-explanatory, therefore please add written out abbreviations to the legends where missing. I do not understand why single words, such as (serum) ferritin (SF), ferroportin (FPN) and hyperferritinemia (HF) are abbreviated throughout the article. This does not improve readability at all...
Author Response
This is a comprehensive and clearly written non-systematic review of potential causes and effects of hyperferritinemia, including guidelines for diagnostics and follow-up.
We are grateful for these comments.
-In the abstract and 1st paragraph, 'low SF is highly specific for iron deficiency anaemia': this should just be 'iron deficiency' as you are not referring to any hemoglobin measurement,
This is corrected.
-In Figure 2, it is not clear what the lines and arrows represent; please add text.
This has been specified in the figure.
-Table 1: what is 'ethnicity associated hyperferritinemia'? Please be more specific.
We have decided to waive this term in the article as it is not well defined in the literature.
-Page 6/7: 'world pandemic' is a pleonasm. It is then stated that 'all patients with COVID-19 should be screened for hyperinflammation', which does not make sense for mildly symptomatic cases.
This is corrected.
-Figure 5: please add definitions of hyperferritinemia, anemia (which is misspelled as anaemia here) etc. For consistency: instead of 'rule out...' in some of the boxes, delete those words and add yes/no to the arrows coming from those boxes.
This is corrected and further adjustments have been made for Figure 5.
-Table 3: how does this relate to Figure 5?
Table 3 is a continuation of Figure 5 regarding indication for MRI, but for simplicity we have decided to keep them separate.
-Page 19: 'Patients with mildly increased SF (300-1000 μg/L) where iron overload is unlikely and other causes are not evident, should be given lifestyle advice such as alcohol abstinence, improved glycaemic control, weight reduction and diet instruction to lower triglyceride concentrations.' This does not make sense, unless these lifestyle behaviors have been found as a potential cause.
We have specified this statement (Lines 604-607).
-Abbreviations: figures should be self-explanatory, therefore please add written out abbreviations to the legends where missing. I do not understand why single words, such as (serum) ferritin (SF), ferroportin (FPN) and hyperferritinemia (HF) are abbreviated throughout the article. This does not improve readability at all...
We have decided to avoid such abbreviations throughout the article.